# Auditory Event-Related Potentials in Two Rat Models of Attention-Deficit Hyperactivity Disorder: Evidence of Automatic Attention Deficits in Spontaneously Hypertensive Rats but Not in Latrophilin-3 Knockout Rats

**DOI:** 10.3390/genes16060672

**Published:** 2025-05-30

**Authors:** Logan M. Brewer, Jankiben Patel, Frank Andrasik, Jeffrey J. Sable, Michael T. Williams, Charles V. Vorhees, Helen J. K. Sable

**Affiliations:** 1Department of Psychology, 202 Psychology Building, University of Memphis, Memphis, TN 38152, USA; lbrewer4@memphis.edu (L.M.B.); janki.pa99@gmail.com (J.P.); fndrasik@memphis.edu (F.A.); 2Department of Behavioral Sciences and Neuroscience Program, Christian Brothers University, Memphis, TN 38104, USA; jsable@cbu.edu; 3College of Medicine, University of Cincinnati, Cincinnati, OH 45220, USA; michael.williams@cchmc.org (M.T.W.); charles.vorhees@cchmc.org (C.V.V.); 4Cincinnati Children’s Research Foundation, Cincinnati, OH 45229, USA

**Keywords:** ADHD, SHR, *Lphn3*, auditory-evoked potential, auditory processing, animal, genetic models

## Abstract

Background/Objectives: Variations of the latrophilin-3 (*Lphn3*) gene have been associated with attention-deficit hyperactivity disorder (ADHD). To explore the functional influence of this gene, *Lphn3* knockout (KO) rats were generated and have thus far demonstrated deficits in ADHD-relevant phenotypes, including working memory, impulsivity, and hyperactivity. However, inattention remains unexplored. Methods: We assessed automatic attention in *Lphn3* KO (n = 19) and their control line (wildtype/WT, n = 20) through use of the following auditory event-related potentials (ERPs): P1, N1, P2, and N2. We also extended this exploratory study by comparing these same ERPs in spontaneously hypertensive rats (SHRs, n = 16), the most commonly studied animal model of ADHD, to their control line (Wistar–Kyoto/WKY, n = 20). Electroencephalograms (EEG) were recorded using subdermal needle electrodes at frontocentral sites while freely moving rats were presented with five-tone trains (50 ms tones, 400 ms tone onset asynchronies) with varying short (1 s) and long (5 s) inter-train intervals. Peak amplitudes and latencies were analyzed using GLM-mixed ANOVAs to assess differences across genotypes (KO vs. WTs) and strains (SHRs vs. WKYs). Results: The KOs did not demonstrate any significant differences in peak amplitudes relative to the WT controls, suggesting that the null expression of *Lphn3* does not result in the development of inefficiencies in automatic attention. However, the SHRs exhibited significantly reduced peak P1 (and peak-to-peak P1–N1) values relative to the WKYs. These attenuations likely reflect inefficiencies in bottom-up arousal networks that are necessary for efficient automatic processing. Conclusions: Distinct findings between these animal models likely reflect differing alterations in dopamine and noradrenaline neurotransmission that may underlie ADHD-relevant phenotypes.

## 1. Introduction

### 1.1. ADHD and Lphn3

Attention-deficit hyperactivity disorder (ADHD) is a neurodevelopmental disorder characterized by hyperactivity, disinhibition, and inattention that interferes with an individual’s daily life [1]. The national prevalence of ADHD has been estimated at 11.4% in children and adolescents [2] and 6% in adults [3], with functional impairments persisting into adulthood for upwards of two-thirds of children diagnosed [4]. The Diagnostic and Statistical Manual (DSM-V-TR) recognizes three presentations of ADHD: inattentive, hyperactive/impulsive, or a combination of the two [1]. However, individuals usually present with highly heterogenous and multifaceted symptoms that may change over the course of development and do not necessarily fit within these presentations (for a review, see [5]). This complexity can make the treatment and management of symptoms difficult. Therefore, furthering our understanding of its etiology and pathophysiology may aid in better differentiating subsets of presentations and treating individuals with diverse profiles.

ADHD is highly heritable, with a reported mean heritability of 74% [6]. In addition, it is generally accepted that symptom dysfunction falls along a dimensional continuum, with those with “true” ADHD representing the extreme end of it [7,8]. This is supported by both twin and molecular studies [9]. Thus, the delineation of risk genes that contribute to the etiology and pathophysiology of ADHD may prove valuable to understanding the disorder. Variants of the latrophilin-3 (*Lphn3*), or adhesion G protein-coupled receptor L3 (*Adgrl3*), gene have been shown through genome-wide linkage [10,11,12] and association studies [12,13] to be associated with ADHD. Further, a recent meta-analysis of the most common single-nucleotide polymorphisms (SNPs) of *Lphn3* reported significant associations between rs6551665 and rs2345039 SNPs and the presence of childhood ADHD [14]. Thus, understanding the functional significance of *Lphn3* in ADHD may advance our knowledge of its neurobiology.

Notably, substantial genetic overlap exists with other externalizing, internalizing, and neurodevelopmental symptoms (for reviews, see [6,15]). However, this overlap is rarely addressed, in part due to the common practice of comparing groups within the conceptual bounds of DSM criteria (i.e., those with “ADHD” versus controls). Therefore, to better delineate the neurobiological underpinnings of dysfunctional phenotypes across and within different disorders, we adhere to the framework outlined by the research domain criteria (RDoC) [16,17]. RDoC shift the focus away from polythetic conceptualizations inherent in DSM-based categories to dimensional constructs that are understood to cut across psychiatric disorders (e.g., motivation, disinhibition, inattention, etc.). Furthermore, by moving from modeling complex clusters of human behavior to functional dimensions grounded in neurobiology, animal models can be generated with greater precision [18,19].

### 1.2. Lphn3 and Animal Models of ADHD Phenotypes

Animal models are useful for exploring the “downstream” effects of relevant genes on the neurobehavioral mechanisms that underlie mental disorders [18]. To understand the function of *Lphn3*, a growing number of genetic knockdown (KD; i.e., subjects that have reduced expressions of *Lphn3*) and knockout (KO; i.e., subjects that are null for the expression of *Lphn3*) models have been produced and studied, including Drosophila KDs [20], zebrafish KDs [21,22,23,24], and mouse KOs [25,26,27]. Although these models have contributed to advancing our understanding of the function of *Lphn3*, they lack the capacity to assess cognitive domains commonly associated with ADHD (e.g., working memory, impulsivity, learning, and memory).

To address this limitation, an *Lphn3* KO rat was developed [28]. These rats have been found to be hyperactive in both novel and—unlike KO mice [27]—familiar environments [28]. Additionally, they are impulsive, demonstrate working memory inefficiencies [29,30,31], and exhibit selective deficits in egocentric and allocentric learning and memory [32]. However, despite inattention being a core characteristic of some presentations of ADHD, it has been assessed only indirectly in this model using a non-cued alteration task [31]. Thus, it remains an open question whether *Lphn3* KOs exhibit deficits in attention.

### 1.3. Auditory-Evoked Potentials and Attention in ADHD

Event-related brain potentials (ERPs) are the summation of neural activity time locked to the presentation of a stimulus and recorded using electroencephalography (EEG) [33]. The waveform produced contains various peaks that are denoted based on their voltage polarity (“P” for positive and “N” for negative) and timing, with the latter either defined by its latency in milliseconds (e.g., P50) or the order it occurs in the waveform (e.g., P1 is the first positive peak). These peaks correspond to components that are associated with either exogenous (i.e., obligatory responses related to the characteristics of the eliciting stimulus) or endogenous (responses related to information processing and characteristic of the subject) properties, with the latter components typically having more endogenous influences [33]. Importantly, ERPs have clear genetically determined differences in humans e.g., [34], as well as rodent models of psychopathology e.g., [35], and neurodevelopment e.g., [36]. Additionally, rats exhibit relatively homologous patterns in ERP components in response to passive visual and auditory stimuli e.g., [37,38,39], to those of humans, making them a viable translational tool.

A subset of ERPs, auditory-evoked potentials (AEPs), allows for the examination of auditory processing at distinct levels of the auditory system from the cochlea to the cortex [40]. AEPs are organized into three categories based on the latency from stimulus onset: brainstem auditory-evoked potentials (BAEPs; 1–10 ms), middle latency auditory-evoked potentials (MLAEPs, 10–50 ms), and long latency auditory-evoked potentials (LLAEPs, 50–300 ms). LLAEPs are primarily measured from the scalp/dorsal cephalic position (referenced to post-auricular positions) and are colloquially referred to as ERPs. They consist of two obligatory components, the positive P1 and the negative N1, followed by alternating positive and negative peaks that reflect activity beyond primary auditory processing and are not specific to the stimulus (for a review, see [40]).

Of these components, the auditory N1 (or N100) is primarily exogenous, as it is predominantly influenced by the characteristics of the presented stimulus (e.g., intensity, saliency). However, it also has endogenous properties. Specifically, the N1 is modulated by both arousal (i.e., sensitivity to detecting environmental stimuli) and attention (i.e., allocation of cortical resources to stimuli), with a more robust N1 observed when individuals are physiologically alert and actively attending to a stimulus (for reviews, see [41,42]). In short, the N1 may function as a neural correlate of the sensory processing of an eliciting stimulus that is further modulated by one’s selective attention (i.e., top-down voluntary processes) and arousal (i.e., bottom-up automatic processes).

Given that attention may modulate the auditory N1, it is reasonable to predict that its presentation may be altered in those with ADHD; however, findings have so far been mixed. While some have reported that the N1 is generally attenuated in those with ADHD (for reviews, see [43,44]), a more recent meta-analysis reported no significant differences between ADHD and non-ADHD participants [45]. That said, these inconsistencies may be explained by several limitations, including an inability to account for several relevant moderators (e.g., specific ADHD presentation, comorbidities, symptom severity), a sole focus on males, and the fact that most of the reviewed studies feature populations that consist of children ages 6–11 years old. Additionally, this meta-analysis focused on ERPs with prominent cognitive components elicited by active tasks (e.g., selective attention, inhibitory control, and error monitoring) [45]. This is notable given that effortful attention may produce larger increases in the N1 in those with ADHD [46,47], while the N1 is reduced in the context of automatic, non-effortful attention [48].

Sable et al. [48] examined the association between automatic attention and the N1 in young adults with and without ADHD (based on self-reported diagnosis from a physician) using a passive auditory paradigm. Participants were asked to ignore auditory tone trains separated by either a short (1 s) or long (5 s) inter-train interval (ITI) while watching a silent video. As anticipated, the control group demonstrated a more robust N1 following the longer, more salient ITI; however, this difference was absent in adults with ADHD. This attenuation of the N1 was understood to reflect deficits in automatic attention in those with ADHD, with this deficit being primarily driven by a lack of bottom-up gain from subcortical structures [48].

### 1.4. The Current Study

To understand the influence of *Lphn3* on inattention, we examined the auditory-evoked P1 and N1 and other auditory ERPs (P2 and N2) in *Lphn3* KO rats compared with their wildtype (WT) control line [28]. As it is not fully understood whether the P1 or N1 is more analogous with the human N1 [49], we focused on changes in both components. To accomplish this, we utilized a modified version of the auditory paradigm previously implemented in humans to explore automatic attention [48]. This passive auditory paradigm uses trains of five tones with inter-tone intervals (onset to onset) of 400 ms, each separated by either a 1 s (short) or 5 s (long) ITI. By varying the period of silence between each train, it is possible to examine both the attenuation of AEPs across tones (i.e., a reduction in response across repeated tones in train) and the recovery of the response (i.e., a greater response following the 5 s ITI than 1 s ITI), which can take as long as ~5 s in rats [50]. Therefore, this paradigm allows for the examination of not only the absolute P1 and N1 amplitudes, respectively, but also the interaction between attenuation (i.e., inhibition) and recovery (i.e., release from inhibition; see [51] for supporting information on the distinction between these processes).

We also assessed these auditory ERPs in a more established animal model of ADHD, the spontaneously hypertensive rat (SHR). The SHR line was created by selectively breeding Wistar–Kyoto (WKY) rats from disparate lines for high cytosolic blood pressure [52]. SHRs have demonstrated good face validity, with evidence of hyperactivity, impulsivity, and inattention (for a recent review, see [53]). Notably, however, only sustained attention has been evaluated in this model [54,55]. While a single study was found that noted altered sensory gating based on ERP changes in SHRs [56], automatic attention processes have remained broadly unexplored.

In summary, we aimed to determine whether *Lphn3* KO and SHRs demonstrate deficits in automatic attention, as indicated by reductions in P1 and/or N1 amplitudes to the first tone following the 5 s ITI. By doing this, we sought to determine whether the null expression of *Lphn3* produces deficits in automatic attention, while adding to the construct validity of the SHRs.

## 2. Materials and Methods

### 2.1. Animals

All procedures were sanctioned by The University of Memphis Institutional Animal Care and Use Committee (IACUC; protocol #0864) and conducted in accordance with the Public Health Service Policy on Humane Care and Use of Laboratory Animals [57]. A total of 21 SHRs (9 F and 12 M) and 24 WKYs (12 F and 12 M) were obtained from Charles River Laboratories (Kingston, NY, USA) and arrived at the University of Memphis at approximately postnatal day (P) 45. A total of 22 *Lphn3* KOs (11 M and 11 F) and 24 WTs (14 F and 10 M) from a Sprague–Dawley background (SD-IGS, strain 001, Charles River Laboratories, Charleston, NC, USA) were acquired from the Cincinnati Children’s Research Foundation at approximately P30. The KOs were generated at the Cincinnati Children’s Transgenic Animal and Genome Editing Core utilizing CRISPR/Cas9 to delete exon 3 (see [28] for details regarding this process and the confirmation of the successful deletion of *Lphn3*). One rat per genotype per sex per litter was randomly selected for this experiment with the use of a random number table.

Upon delivery, same-sex and same-strain rats were pair-housed in standard plastic cages (45 cm × 24 cm × 30 cm) with corn cob bedding. They were then housed in a room with controlled temperature (23 ± 2 °C) and humidity (30–70%) and a 12 h reverse light/dark cycle (lights off at 0700 h). Rats had ad lib access to feed (Teklad 2018, Lafayette, IN, USA) and water until P60, at which time they were placed on a food restriction schedule and maintained at 85–90% of free-feeding weight, which was adjusted bi-weekly to account for any growth. Behavioral testing began at P70 (see [31] for details). At the conclusion of behavioral testing, rats remained pair-housed and were transitioned to free feed for the duration of this study. The average postnatal (P) age of the KO rats was P182.33 (SD = 19.79) and the average P age of the WTs was P182.25 (SD = 22.79) at the time of EEG data acquisition, while that of the SHRs was P196.32 (SD = 5.24) and that of the WKYs was P195.63 (SD = 5.36).

### 2.2. Electrode Placement

ERPs were recorded in awake rats via a method we adopted for use in both awake and anesthetized rats [58]. Each rat was initially placed in an induction box and anesthetized with 3–4% isoflurane in oxygen until sedated, as confirmed by the absence of tail and toe (pedal) pinch reflexes. They were then transported to the operating table and placed on an electric heating pad, where anesthesia depth was maintained using 1–2% isoflurane in oxygen via a nose cone. The fur from the scalp to the base of the neck was shaved using an electric razor, and any remaining stubble was removed using a depilatory of potassium thioglycolate. The exposed skin was thoroughly rinsed with water and soaked gauze, swabbed with betadine, and cleaned with a 70% isopropyl alcohol pad.

Three 7 mm disposable subdermal needle electrodes (Rhythmlink; Columbia, SC, USA) were inserted into the scalp. The recording electrode was placed ~3 mm anterior to the bregma, terminating directly above the frontal cortical region (roughly homologous to Fz in humans). The reference electrode was caudal to the right ear, while the ground was symmetrically placed caudal to the left ear. Impedance was checked using Biopac MP36 and BSL software version 4 (Biopac System Inc.; Goleta, CA, USA), with values below 15 kΩ considered acceptable. An initial administration of cyanoacrylate adhesive (i.e., super glue) was used to secure the electrodes and then supplemented by a generous application of Collodion-soaked gauze. Tegaderm HP (3M; St. Paul, MN, USA) transparent dressing was then placed over the electrodes once the collodion had dried. The rat was then outfitted with a jacket (Instech Laboratories Incorporated; Plymouth Meeting, PA, USA) connected to a flexible stainless steel spring to cover and secure the lead wire. Once the jacket was confirmed to be secured, but not excessively tight such that it impeded movement or caused discomfort, the isoflurane flow was discontinued so that the rat was only receiving pure oxygen through the nose cone. After the rat voluntarily removed its nose from the nose cone, it was placed in a recovery chamber for an additional 30 min. Once fully recovered, it was moved to a 50 cm × 50 cm × 25 cm sound-attenuated chamber for ERP recordings. This entire process took approximately 35 to 40 min to complete for each rat.

### 2.3. ERP Recordings and Auditory Stimuli

Once the rat was in the testing chamber, the lead wires were fed through the top and connected to a swivel to permit the rat to freely move throughout the chamber. Each rat was given ~5 min to acclimate to the chamber prior to EEG acquisition. The lead wires were connected to an integrated ERP recording system utilizing a Biopac MP36 and BSL software (Biopac System Inc.; Goleta, CA, USA). EEG data were sampled at 500 Hz with a DC to 100 Hz band pass filter and a 60 Hz notch filter. Data were further filtered offline using an FIR filter with a 0.01–30 Hz bandpass (for further information regarding data collection setup, see [58]).

Following 2 min of recording resting EEG data, the rat was presented with a series of tone trains, each preceded in randomized order by either a 1 or 5 s ITI, for a total of 320 trains/ITIs (160 long and 160 short). The trains consisted of ~90 dB complex tones with a 500 Hz fundamental frequency and harmonics at 1000 and 1500 Hz (−3 dB and −6 dB, respectively, relative to the fundamental). All tones were 50 ms in duration, with 5 ms Gaussian onset and offset periods and 400 ms stimulus onset asynchronies (SOAs) within tone trains (see Figure 1). The tones were calibrated prior to each recording session using an ambient sound meter. Tone trains played through a speaker centered near the top of the back of the testing chamber and were presented using SuperLab version 4.5 (Cedrus Corp., San Pedro, CA, USA). The onset of each ITI was recorded and marked in the EEG data file using a StimTracker (Cedrus, San Pedro, CA, USA).

Rats with ≥60 accepted trials were considered to have a valid number of trials to be included in the analyses. KOs had an average of 121 short and long accepted ITI trials (minimum short and long ITI values were 90 and 91, respectively), and WTs had an average number of 125 short and 126 long accepted ITI trials (minimum of 74 and 76 long and short ITI trials, respectively). No significant difference was found in the number of accepted trials between KOs and WTs for short ITIs [t (37) = −0.515, *p* = 0.609] or long ITIs [t (37) = −0.670, *p* = 0.507]. SHRs demonstrated a significantly greater number of rejected trials than WKYs for both short [t (34) = −8.537, *p* < 0.001] and long ITIs [t (34) = −7.764, *p* < 0.001]. This was predominantly related to a high number of movement artifacts observed in SHRs, likely due to their more hyperactive nature. Given this, a select number of SHRs completed ~1–3 additional recording sessions to obtain a sufficient number of usable trials. This process was also performed with the remaining WKYs to maintain consistency across both strains. For each rat, two sessions were selected to maximize the number of usable trials after a visual inspection of the quality of the waveforms. Weighted averages were then calculated from the selected sessions. SHRs had a combined weighted (n = 18) and non-weighted (n = 3) average of 91 short and 93 long accepted ITI trials (a minimum of 55 long and 62 short ITIs), and WKYs had a combined weighted (n = 14) and non-weighted (n = 10) average of 136 for both short and long accepted ITI trials (a minimum of 95 long and 94 short ITI).

### 2.4. EEG Processing and ERP Analyses

A custom MATLAB (version 9.11) script was used to generate ERPs. The EEG was separated into epochs consisting of 200 ms before to 2000 ms after tone train onset, with the 200 ms pre-stimulus baseline mean subtracted from each point in the epoch. Epochs with voltage peaks exceeding ±100 μVs were excluded from further analyses, while remaining epochs were averaged separately for each genotype/strain (KO, WT, SHR, and WKY) and ITI (1 s and 5 s). This process generated an average ERP waveform for each genotype and strain based on the average response across all tone trains for the 1 s and 5 s ITIs. ERP peaks were identified based on a visual inspection of the grand average ERP waveform for the 5 s ITI across control lines (based on the assumption that the responses to the first tone would be largest under this condition), as well as from previous conventions in rodents and humans (e.g., [38,39]). The following ERP peaks were defined: the most positive point from 0 to 100 ms for P1, the most negative point from 50 to 150 ms for N1, the most positive point from 100 to 300 ms for P2, and the most negative point from 200 to 350 ms for N2. The latency range used was wider than previous conventions to better capture each ERP peak across rats, as peaks were more delayed in response to the tone trains than what we observed for tone pairs [58].

Prior to analysis, individual subjects that were judged to have a poor signal quality based on a visual inspection or an insufficient number of usable trials (≤60) were removed. This resulted in the exclusion of 3 KOs (2 males and 1 female), 4 WTs (4 females), 4 SHRs (2 males and 2 females), and 3 WKYs (1 male and 2 females). Following this process, outliers were assessed for peak amplitudes and latencies. Rats with significant outliers (i.e., ±3 SD) across several ERP components and multiple variables were removed, as these outliers were deemed too extreme to be addressed through mean imputation. This resulted in the removal of a 1 additional male SHR and WKY, leaving 19 KO rats (9 males and 10 females), 20 WT rats (10 females and 10 males), 16 SHRs (7 females and 9 males), and 20 WKY rats (10 females and 10 males). Due to the high number of variables, any remaining outliers with 1–2 measures were retained and adjusted using 2 SD imputation with matched between-subject variables (i.e., genotype/strain and sex). This process was performed to maximize power by limiting the removal of rats that contained outliers on a few select measures for one component, while addressing the remaining outliers through mean imputation to reduce their influence on the parametric analyses. Based on a prior power analysis using G*Power version 3.1.9.7 [59], we determined we had adequate samples of KOs/WTs (N = 39) and SHRS/WKYs (N = 36) to detect medium effects (f = 0.25) for mixed GLM ANOVAs, given a significance criterion of α = 0.05 and power = 0.80.

Average peak amplitudes and latencies were analyzed for P1, N1, P2, and N2. Special attention was given to the P1 and N1 peaks, since it is unclear which component is homologous to the human N1 [49], and these components have been observed to most significantly reflect physiological changes in response to a similar tone pair paradigm [58]. Peak amplitudes and latencies were analyzed using two separate 2 (genotype or strain) × 2 (sex: male/female) × 2 (ITI: 1 s/5 s) × 5 (tone: 1–5) mixed GLM ANOVAs for KOs and SHRs, respectively, and their respective control lines (i.e., WTs and WKYs). In addition, peak-to-peak amplitudes were analyzed, as the latter provides an alternative measure of peak amplitude that is not confounded by residual noise or artifacts contained in the pre-stimulus baseline [33]. Peak-to-peak measures were calculated by subtracting the first peak amplitude (e.g., P1) from the second adjacent peak (e.g., N1) and recording the absolute value. This generated three peak-to-peak measures: P1–N1, N1–P2, and P2–N2. Significant or approaching omnibus findings (*p* ≤ 0.05) were followed up using simple pairwise comparisons with Bonferroni adjustments to determine the nature of the significant interactions across various conditions. A Greenhouse–Geisser correction was used for omnibus findings when appropriate to account for violations of sphericity. All analyses were conducted using SPSS version 27.0 (IBM Corp; Armonk, NY, USA). 

Given that no clear sex differences emerged, and for the sake of brevity, we have opted to limit the reported results to peak amplitudes and latencies collapsed across sex [see Appendix B for significant peak-to-peak and sex statistical findings and the Appendix A for inferential statistics on *Lphn3* KO versus WT (Appendix A) and on SHRs versus WKYs (Appendix A)].

## 3. Results

### 3.1. Lphn3 KO Rats

Grand average waveforms for KOs and WTs are shown in Figure 2. Separate waveforms are presented for each genotype and ITI collapsed across sex. The waveform demonstrated the anticipated pattern of a larger response to the initial tones of the trains, with this response being most prominent following the longer ITI. This pattern resembled those found when a similar tone pair paradigm was used in awake Wistar rats [58], as well as what has been previously demonstrated in humans using this paradigm e.g., [48,60,61]. However, the P2 and the N2 were notably not as robust, with both being almost absent following the shorter ITI.

#### Peak Amplitudes and Latencies in KOs Versus WTs

All peak amplitude and latency inferential statics for KOs and WTs are provided in Appendix A. The peak amplitudes for each ERP component, collapsed across sex, are depicted in Figure 3. The P1 amplitude (shown in Figure 3A) was substantially greater following the first tone compared to the second tone [main effect of tone, F (2.64, 92.25) = 35.48, *p* < 0.001, $\eta_{p}^{2}$ = 0.198). The P1 was larger for the first tone following the 5 s ITI compared to the 1 s ITI (a 3.10 µV difference), as well as for the second tone (a 2.14 µV difference; ITI x tone, F (3.069, 107.405) = 18.372, *p* < 0.001, $\eta_{p}^{2}$ = 0.344). No effects of genotype on P1 peak amplitude were found. For peak latency, the P1 peaked earlier following the 5 s ITI (52. 48 ± 2.35 ms) compared to the 1 s ITI (60.73 ± 2.68 ms; ITI, F (1, 35) = 14.956, *p* < 0.001, $\eta_{p}^{2}$ = 0.299), but no significant effects of genotype were seen.

The N1 amplitudes are shown in Figure 3B. Like the P1, the N1 was largest for the first tone, followed by a significant decrease across subsequent tones in the train, with this decrease being more prominent following the longer ITI. The N1 was greater for the first tone when it was followed by the 5 s ITI (−11.87 ± 1.06 µV) compared to the 1 s ITI (−5.43 ± 0.66 µV), with this effect persisting to the second tone in the train (−2.98 µV difference; ITI x tone, F (1.976, 69.165) = 26.924, *p* < 0.001, $\eta_{p}^{2}$ = 0.435). However, no genotype differences were seen in the N1 peak amplitude. Analysis of N1 peak latency revealed a significant sex x tone interaction (Appendix A), but no effects of genotype or ITI emerged.

The P2 amplitudes (see Figure 3C) to the initial tone were less robust than for N1 amplitudes and did not return to baseline (refer to Figure 3B), which resulted in negative values to the first and second tones. The P2 was larger for the first tone following the 5 s ITI compared to the 1 s ITI in both KOs (−4.66 ± 1.08 µV vs. −0.82 ± 0.76 µV) and WTs (−2.70 ± 1.05 µV vs. −0.42 ± 0.73 µV), with this effect persisting to the second tone only in the WTs (−1.53 µV difference). In addition, the P2 significantly decreased from the first tone to the second tone only after the 5 s ITI in KOs (3.09 µV difference) and not the WTs [genotype x ITI x tone, F (2.486, 87.013) = 3.919, *p* < 0.016, $\eta_{p}^{2}$ = 0.101]. The P2 generally peaked earlier following the 1 s ITI (182.90 ± 9.09 ms) compared to the 5 s ITI (200.99 ± 8.01 ms; main effect of ITI, F (1, 35) = 10.958, *p* = 0.002, $\eta_{p}^{2}$ = 0.283). It also peaked earliest for the first tone of the train (175.44 ± 9.30 ms), which was significantly earlier than the second tone (35.10 ms difference; main effect of tone, F (1, 35) = 3.635, *p* = 0.008, $\eta_{p}^{2}$ = 0.094), No genotype-related differences were seen in P2 latency.

Like the P1 and N1 amplitudes, the N2 amplitudes (shown in Figure 3D) were larger for the first and second tones following the 5 s ITI (−8.79 ± 1.04 µV and −5.351 ± 0.59 µV) compared to the 1 s ITI (−5.79 ± 0.56 µV and −3.89 ± 0.43 µV), followed by a decrease in amplitude across subsequent tones, particularly after the 5 s ITI [ITI x tone, F (2.464, 86.234) = 14.131, *p* < 0.001, = 0.288]. In addition, when collapsed across ITIs, the N2 significantly decreased from the first to the second tone in KOs (−3.46 µV difference; genotype x tone, F (1.590, 55.662) = 3.538, *p* = 0.046, $\eta_{p}^{2}$ = 0.092), while this reduction was absent in WTs. However, the difference between KOs and WTs in N2 amplitude for the first tone did not reach the criterion for significance (*p* = 0.054). For N2 peak latency, while a significant main effect of tone was found [F (4, 140) = 3.411, *p* = 0.011, $\eta_{p}^{2}$ = 0.092], no differences between tones in the train were detected following pairwise comparisons (*p* > 0.05). Likewise, no genotype differences or other notable effects emerged for N2 peak latency.

### 3.2. Spontaneously Hypertensive Rats

Grand average waveforms for the SHRs and WKYs, for each strain and ITI, are shown in Figure 4. The waveform for the WKY controls demonstrated a similar pattern as those for the KOs and WTs, with a clear P1 and N1 and less robust P2 and N2. Notably, however, the P1 and N1 were substantially attenuated in the SHRs, with these rats also having a more pronounced N2.

#### Peak Amplitudes and Latencies in SHRs Versus WKYs

All peak amplitude and latency inferential statics for SHRs and WKYs are provided in Appendix A. The peak amplitudes for each individual ERP component, collapsed across sex, are depicted in Figure 5. The P1 amplitudes are shown in Figure 5A. The P1 for the first tone in the train was significantly smaller in the SHRs than in the WKYs, with this reduction being prominent following the 5 s ITI (4.57 µV difference; strain x ITI x tone, F (2.824, 90.382) = 2.996, *p* = 0.038, $\eta_{p}^{2}$ = 0.086). For peak latency, the P1 peaked earliest for the first tone of the train (50.56 ± 2.89 ms) relative to the second (17.19 ms difference) and third tones (11.66 ms difference; main effect of tone, F (4, 128) = 5.365, *p* < 0.001, $\eta_{p}^{2}$ = 0.144); however, no difference in strain was detected.

The N1 amplitudes are shown in Figure 5B. The N1 was larger for the first tone following the 5 s ITI (−4.20 ± 0.61 µV) compared to the 1 s ITI (−1.94 ± 0.41 µV) in the WKYs, while this difference was absent in SHRs (−0.05 µV difference; strain x ITI x tone interaction, F (3.052, 97.670) = 3.591, *p* = 0.016, $\eta_{p}^{2}$ = 0.101)). In addition, the N1 was larger for the fifth tone following the 1 s ITI in the SHRs than in the WKYs (−2.91 ± 0.56 µV vs. −1.94 ± 0.42, respectively). Regarding peak latency, the N1 peaked significantly later for the first tone (116.20 ± 4.58 ms) compared to the third (84.27 ± 5.95 ms), fourth (89.43 ± 4.49 ms), and fifth tones (88.71 ± 4.10 ms), but not the second tone (94.38 ± 5.95 ms; main effect of tone, F (4, 128) = 9.856, *p* < 0.001, $\eta_{p}^{2}$ = 0.235). Furthermore, the N1 peaked earlier in the WKYs (87.19 ± 4.07 ms) than in the SHRs (101.99 ± 4.55 ms; main effect of strain, F (1, 32) = 5.868, *p* = 0.021, $\eta_{p}^{2}$ = 0.155).

The P2 amplitudes are shown in Figure 5C. The P2 was not as robust as the prior N1 peak for both WKYs and SHRs (refer to Figure 5B). This appears to be due to the temporal overlap of the N1 and P2 components (i.e., a larger N1 following the first tone and signal regression to the mean), which appeared to be driven predominantly by females across strains (refer to Section A.4). No effects of strain were detected on P2 amplitude. For P2 latency, it peaked earlier for the first tone (129.61 ± 4.13 ms) compared to any subsequent tones in the train (main effect of tone, F (3.115, 99.669) = 7.005, *p* < 0.001, $\eta_{p}^{2}$ = 0.180). No other effects were found for P2 peak latency.

The N2 amplitudes are shown in Figure 5D. The N2 was greater for the first tone followed by the 5 s ITI (−6.73 ± 0.48 µV) compared to the 1 s ITI (−5.16 ± 0.33 µV), and it was attenuated across subsequent tones (ITI x tone, F (2.252, 72.076) = 7.710, *p* < 0.001, $\eta_{p}^{2}$ = 0.194). Notably, the N2 was larger in the SHRs than it was in the WKYs for the first (−3.19 µV difference), second (−2.03 µV difference), third (−2.11 µV difference), and fifth tones (−1.69 µV difference; strain x tone, F (2.694, 86.195) = 4.551, *p* = 0.006, $\eta_{p}^{2}$ = 0.127). The N2 peak latency for the first tone occurred earlier when it was preceded by the 5 s ITI (255.70. ± 6.21 ms) compared to the 1 s ITI (281.61 ± 8.2 ms; ITI x tone, F (3.440, 110.073) = 2.689, *p* = 0.034, $\eta_{p}^{2}$ = 0.078), but no significant effect of strain was detected.

## 4. Discussion

The overall ERP waveform demonstrated the anticipated pattern. Excluding the SHRs, the first tone of the tone train following the 5 s ITI elicited a robust P1–N1 complex, followed by the attenuation of responses to the subsequent tones across KO, WT (see Figure 2), and WKY rats (see Figure 4). Several processes may explain this: (1) the greater salience of the first tone of the train following the longer ITI elicited greater neural activation of perceptual and attentional networks, (2) latent inhibition from frontal inhibitory networks reduced subsequent responses in the train [61], or (3) a combination of these two processes.

Additionally, differential effects were seen across these two rat models. While no meaningful effect of genotype was observed for any of the assessed ERPs between the *Lphn3* KOs and their WT controls, the SHRs exhibited a very clear attenuation of P1 (see Figure 5A) and P1–N1 (see Figure A1 and Figure A2) relative to the WKYs.). Furthermore, the N2 was more pronounced in the SHRs than the WKY rats. While source localization could not be delineated given the use of a single channel, this suggests that inputs from more cortical areas increased as the signal propagated from subcortical to cortical areas, which may function as a compensatory mechanism in SHRs. Together, these differences in ERP peaks suggest that SHRs exhibit deficits in automatic attention, whereas *Lphn3* KOs do not. This likely reflects differences in the underlying neurobiology of the attentional networks in these two rat models.

### 4.1. SHRs and the Locus Coeruleus

Attention encompasses a range of neural networks that engage both sensory and cognitive processes. One theory proposed by Posner and Petersen [62,63], divides attention into three relatively distinct networks: (1) alerting (acquiring and maintaining a level of tonic arousal needed for sustaining vigilance), (2) orientating (prioritizing and directing attention to sensory stimuli), and (3) executive functioning/focal attention (e.g., top-down control of sensory stimuli). Within this framework, attentional issues in ADHD have been suggested to result from dysfunction in both the executive and alerting networks [64]. Automatic attention is under the influence of both the frontal cortical (executive) and subcortical (alerting) networks, with the former functioning to inhibit irrelevant stimuli and the latter sustaining optimal arousal/alertness to detect salience and changes in the environment [65].

It has been suggested that N1 attenuation to repeated stimuli in humans is a consequence of top-down inhibitory mechanisms that exert habituating effects on increasingly irrelevant sensory stimuli [61]. This is supported by the reduced N1 attenuation in older adults, given that decreased activity in these frontal inhibitory networks occurs with aging [60]. However, the reduced amplitude of the P1/P1–N1 in the SHRs was primarily driven by a substantial attenuation to the first tone of the tone train, particularly after the 5 s ITI. This suggests that inefficiencies in bottom-up subcortical arousal networks, rather than in frontal top-down inhibitory networks account for this reduction in N1 amplitude. This pattern is similar to what was observed in young adults with ADHD using a similar version of this paradigm [48].

Halperin and Schulz [66] proposed that ADHD is a condition that reflects dysfunction in relatively developmentally static subcortical structures, with the maturation of the prefrontal cortex and white matter tracts leading to reductions in ADHD symptoms with age [4]. This proposal implicates several systems and regions, possibly including the basal ganglia/midbrain dopamine (DA) system, the hindbrain norepinephrine (NE) system, and the cerebellum [66]. Of these, the NE system originating from the locus coeruleus (LC) is a likely candidate for underlying the attenuation of the P1 and P1–N1 complex in the SHRs. 

The LC, as well as sub-coeruleus α1 and α2 brain stem nuclei, serve as the primary neural substrates of NE [67,68]. Although the LC-NE system has been implicated across an array of cognitive processes (for reviews, [65,69,70,71,72]), one function it serves is the maintenance of an organism’s level of arousal (i.e., the ability to receive information) and activation (i.e., the ability to react to information) to effectively respond to a given stimulus [73]. This occurs through two broad processes: (1) longer, slower tonic shifts that function to adjust or maintain a level of arousal best suited for responding to a particular context, such as during sleep versus scanning for threats [67], and (2) shorter, faster phasic firing in response to a stimulus, particularly task- or goal-related stimuli [73], with salient stimuli eliciting a more pronounced phasic response [74]. For most individuals, tonic firing can efficiently adjust to address the need for focused or flexible attention, while these dynamic shifts may be impaired in those with ADHD or ADHD symptoms [75].

Aboitiz et al. [76] proposed that an imbalance in tonic catecholaminergic signaling, particularly DA and NE, leads the attentional network in those with ADHD to be less effective at adjusting to meet environmental demands. Specifically, dysfunctions in these tonic shifts are believed to contribute to dysregulated phasic activation, with supra-optimal tonic levels leading to a poorer “signal-noise ratio” for salient information and bursts of impulsive responding and super-optimal levels leading to oversensitive phasic threshold activation and distractibility. This proposal aligns with the cognitive–energetic model of ADHD, which views ADHD as stemming from a deficit of response readiness and inhibition due to impairments in these energetic pools from bottom-up subcortical areas [77,78] (also, see [79] for an introduction to the cognitive–energetic model).

SHRs have higher concentrations of NE in both the prefrontal cortex and LC [80,81], as well as hypernoradrenergic and hypodopaminergic activity in the prefrontal cortex (PFC) compared with controls [81]. Additionally, unlike Wistar rats, functional α1 adrenergic receptors involved in increasing the release of NE remain active in the LC neurons of SHRs throughout development, which may weaken α2 signaling and impair LC function [82]. These findings jointly suggest that SHRs possess an imbalance in DA and NE similar to that seen in some individuals with ADHD. This implies that SHRs have inefficient tonic and phasic LC activation, a notion supported by evidence that the sensory gating of auditory ERPs is selectively influenced by arousal in frontal and parietal areas in the rat brain [83], with high [84] and low tonic levels contributing to the attenuation of these ERPs [85]. In addition, the increased N2 in SHRs may reflect a compensatory mechanism in cortical activation to compensate for this reduced bottom-up input due to the imbalance in LC signaling, but this idea requires further investigation.

### 4.2. Lphn3 KOs and Dopamine Dysfunction

*Lphn3 KO* rats exhibited similar ERP peaks as their WT counterparts (Figure 2). This appears to reflect the preservation of automatic attention and, possibly, the alerting attentional network in this model, unlike in the SHRs. Subsequently, this null finding in the context of the noted deficits in other ADHD-relevant behaviors [27,28,29,30,31] suggests relevant differences in neurobiology between these two ADHD animal models.

LPHN3 is predominantly expressed in the brain [86,87], with the highest levels found in the PFC, caudate nucleus, amygdala, and cerebellum [11]. It is believed to play a role in synaptogenesis and synaptic connectivity [88,89,90], and its expression decreases throughout development [11]. *Lphn3* KO rats exhibited no differences in the levels of DA, NE, serotonin, or other major metabolites in the striatum, hippocampus, or PFC compared with WT rats. However, tyrosine hydroxylase (TH) and aromatic L-amino acid decarboxylase were elevated. This was understood to indicate a higher availability of DA and/or NE in the striatum [28]. Additionally, KOs exhibited increased striatal DA transporter (DAT) density [28], as well as saturated phasic DA neurotransmission in the neostriatum (caudate–putamen; [91]) relative to WTs, which was suggested to partially account for the downregulation of D1 receptors [28]. More recently, Sable et al. [92] reported somewhat discrepant findings, with KOs exhibiting reduced sum phasic DA release and DAT in the nucleus accumbens (NAcc) core and medial prefrontal cortex (mPFC). However, while these differences may be related to methodology, it is also possible that the null expression of *Lphn3* produced differential changes in two separate pathways. Specifically, it is possible that *Lphn3* KO rats possess hyperdopaminergic signaling in the dorsal stream related to stimulus response [91] and hypodopaminergic signaling in the NAcc-mPFC ventral stream associated with motivation and goal-directed behavior [92].

These alterations in DA signaling pathways provide support for the behavioral phenotypes of KO rats. Specifically, increased transient DA release in the neostriatum may contribute to KO rats’ hyperactive behavior, while phasic decreases in DA in the NAcc [92] may underlie their deficit in impulsive action relative to WTs [30,31]. Similarly, it is likely that differences in the mesocorticolimbic pathways between *Lphn3* KOs and SHRs can explain the present null ERP findings between KO and WT rats relative to the profound differences seen for SHRs. For instance, the lower NAcc-to-mPFC DA ratio observed in SHRs relative to WKYs implies that SHRs have more pronounced deficits in coherence between the mesolimbic and mesocortical pathways than KOs when compared with their WT counterparts [92]. These differences are further reflected by the presence of inefficiencies in impulsive choice in SHRs but not KOs [29]. Although it is currently unclear whether a DA/NE imbalance is present or at least functionally similar in *Lphn3* KOs relative to SHRs, these differences in DA signaling and the present findings appear to implicate differences in the disruption of the mesocorticolimbic pathways between these models. We suggest that tonic catecholaminergic signaling, at least in the alerting network, is likely relatively spared in *Lphn3* KOs versus SHRs.

### 4.3. Considerations and Limits

Although the present findings are promising, some factors merit consideration during interpretation. First, it is worth noting that the *Lphn3* KO rat reflects the null expression of *Lphn3*; therefore, it does not provide a direct model of the risk variants of *Lphn3* found in humans with ADHD [11]. However, the *Lphn3* KO rat still provides insights into relevant neurobiological systems that may be implicated (or, in the case of our findings, not implicated) in individuals with risk variants of the gene that could not be explored otherwise.

Second, although the SHR is a well-validated and -supported animal model for examining ADHD-relevant phenotypes, it has limitations. One concern is that SHRs are, foremost, a model of hypertension. Hypertension typically emerges in SHRs as early as 10 weeks [52] and remains stable at 7 months [93]. This can be problematic for auditory studies, given that hypertension can exacerbate age-related hearing loss [94,95], with changes occurring as early as 3 months [93]. This raises validity concerns, as it could be argued that the reductions in auditory ERPs observed in the SHRs reflect reduced inputs from the auditory cortex due to cumulative cochlear damage [95]. Although we did not directly assess hearing loss, the likelihood that the results simply reflect a loss of hearing is unlikely given several factors. First, researchers have found that the rat dorsal-recorded ERPs receive contributions from both the medial and subcortical structures and are not entirely dependent on the auditory cortex [96]. Second, the intensity of the tones used in the experiment was great enough to likely compensate for any loss of hearing based on reported auditory thresholds in this model [93]. Finally, the results are based on the interactions among ERP amplitudes, not on simple main effects between the groups. These considerations help affirm that our findings represent true neurobiological effects that are not confounded by hearing loss. Even so, it is important to remain mindful of and guard against the potential untoward effects of hypertension when investigating this model.

Third, given the hyperactive nature of the SHRs, adjustments had to be made to ensure that an adequate number of trials were obtained to generate reliable ERPs. Our decision to use weighted averaging may have contributed to some artificial reductions in the waveform due to regression to the mean. However, weighted averages were also used to generate ERPs for the majority of WKYs. In addition, the attenuation of the P1 and P1–N1 in the SHRs was so pronounced that it seems unlikely that it simply reflects an artifact from post-processing. While the use of anesthetized versus awake rats may have addressed the issue of movement artifacts, low levels of anesthesia can have a profound effect on auditory-evoked activity from the cortex and lower brain areas [96], with general anesthesia producing a substantial decrease in these ERPs [58]. In the future, SHRs may benefit from extended time to habituate to the testing chamber, as we observed reduced locomotor activity and a greater number of accepted trials for subsequent recordings.

Lastly, it should be noted we used a less typical, semi-invasive EEG acquisition method, with subdermal needle electrodes. A more invasive method, such as epidural screw electrodes, could have provided improved signal quality relative to that provided by the subdermal needle electrodes used. However, our prior work with these subdermal needle electrodes demonstrated that they can be used to acquire valid auditory ERP peaks from awake rats [58]. This less invasive method is important, as it strikes a balance between minimizing harm to the rat and producing reliable data, in accordance with the “Three R’s” of animal research [97].

### 4.4. Clinical Implications and Future Research

The current findings of an attenuated P1/P1–N1 in SHRs and not *Lphn3* KOs, which reflects relevant differences in underlying brain mechanisms, have implications regarding the development of ADHD medications. At present, stimulants, including methylphenidate (MPH), amphetamine (AMP), and dextroamphetamine (D-AMPH), are the current first-line medications for the treatment of ADHD, while a nonstimulant (e.g., atomoxetine) may function as alternative [98]. Broadly, these drugs increase extracellular catecholamine concentrations, particularly DA and NE; however, their methods of modulating DA and NE levels differ mechanistically based on the relative density of DAT and norepinephrine transporter (NET) in different regions of the brain in conjunction with the levels of affinity of DA and NE for these transporters [99]. Therefore, the beneficial effects of these drugs (i.e., the relief of ADHD symptoms) could be dependent on the underlying brain mechanisms. By having a clearer understanding of these brain differences in relation to a medication’s mechanism and regional specificity on ADHD phenotypes, clinicians could more effectively individualize medication regimens.

The use of these auditory ERPs could provide an index to explore the influence of different stimulants and nonstimulants on automatic attention in rat models of ADHD phenotypes. In SHRs, MPH has been shown to improve some aspects of impulsivity and attention, but not hyperactivity [100], while atomoxetine had null effects on impulsive action [101] or choice [102]. As noted, SHRs and *Lphn3* KOs exhibit divergent disruptions in the mesocorticolimbic pathways, which is reflected by the reduced P1/P1–N1 in the SHRs and not the KOs. We have suggested that medications that either selectively decrease DE and NE in mPFC or increase DA and NE in the NAcc may be most effective at addressing the phasic DA disbalance in SHRs [92]. Testing this hypothesis by assessing whether the P1/P1–N1 is normalized in medically treated SHRs presents a unique opportunity to further characterize the brain mechanisms that underlie automatic attention (i.e., arousal) in this model.

Beyond this, our findings highlight the translational value of these auditory ERPs as clinical tools for bridging preclinical and human research. The pattern elicited by these tone trains was similar to that observed in human controls using this paradigm [48,60], which suggests that the neurophysiological mechanisms are being activated in a similar manner in both humans and rats. In addition, the P1/P1–N1 attenuation seen in the SHRs approximated that observed in young adult humans with ADHD [48]. These homologous qualities support the use of these ERPs as an index of automatic attention across species. Additionally, this means that they could be used to study the influence of different medications in rats and human participants alike. Future research may expand on these findings in human studies of ADHD to assess the degree to which different stimulant and nonstimulant medications modulate the amplitude of these ERPs, as well as how different ADHD profiles influence these effects, or lack thereof, based on neuropsychiatric factors.

## 5. Conclusions

The purpose of this study was to determine whether two animal models of ADHD, the *Lphn3* KO rat and the SHR, demonstrate deficits in automatic attention as indexed by alterations in auditory ERPs, particularly the P1 and N1. SHRs exhibited significantly reduced P1s and P1–N1 complexes relative to WKYs in response to tone trains, indicating deficits in automatic attention. The attenuation of the ERPs was primarily driven by the reduced P1/P1–N1 for the first tone of the train, especially following a longer interval. This reduction implicates the inadequate amplification of the subcortical inputs that comprise the alerting network rather than insufficient top-down inhibition. In contrast, we found no evidence of inefficiencies in automatic attention in *Lphn3* KO rats, with KOs demonstrating similar P1 and N1 peaks relative to WTs. This suggests the preservation of subcortical inputs involved in the alerting network and possibly less disrupted tonic DA and NE signaling in *Lphn3* KO rats than in SHRs. Overall, our findings suggest that *Lphn3* does not contribute to deficits in automatic attention and provide further evidence related to the important distinctions between SHRs and *Lphn3* KOs in the mesocorticolimbic pathways.

In the context of the existing behavioral profile of SHRs, the present findings suggest that this rat model reflects the extreme end of the continuum of ADHD-relevant phenotypes. In short, the SHRs can be viewed as the animal model equivalent of humans with more severe ADHD presentations [7,8], or as a combined ADHD presentation based on DSM categorization. Meanwhile, the *Lphn3* KO model represents the lesser extreme of the functional continuum, reflecting a more impulsive presentation, as indicated by the ADHD-relevant phenotypes of hyperactivity [28] and selective aspects of impulsivity [29,31], while preserving automatic attention.

## Figures and Tables

**Figure 1 genes-16-00672-f001:**
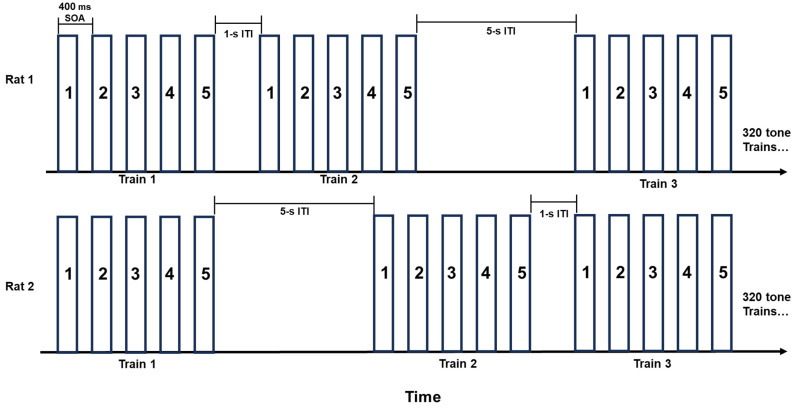
Outline of 5-tone train paradigm. Trains were separated randomly by either 1 or 5 s of silence (offset to onset). Each rat was presented with 320 tone trains, with 160 being preceded by a 1 s ITI (i.e., short) or a 5 s ITI (long). Tones were 50 ms in duration with a 400 ms SOA.

**Figure 2 genes-16-00672-f002:**
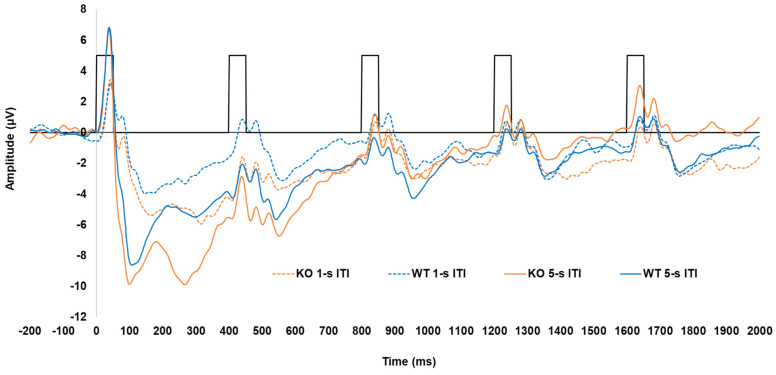
Average ERP waveforms for KOs and WTs to 5-tone trains following both the 5 s (i.e., long) and 1 s (i.e., short) ITIs. Black hollow rectangles indicate the presentation of each 50 ms tone in the train.

**Figure 3 genes-16-00672-f003:**
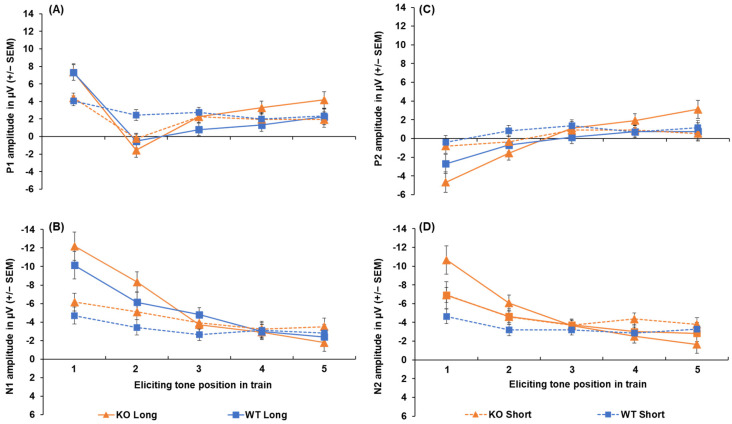
Peak amplitudes of P1 (**A**), N1 (**B**), P2 (**C**), and N2 (**D**) for both *Lphn3* KOs and WT controls not separated by sex for both 1 s and 5 s (long) ITIs. Error bars represent the standard error of the mean (SEM).

**Figure 4 genes-16-00672-f004:**
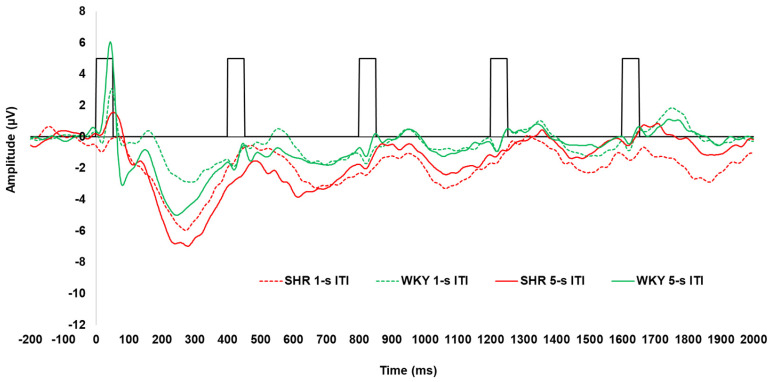
Average ERP waveforms for SHRs and WKYs to a 5-tone train following both the 5 s (i.e., long) and 1 s (i.e., short) ITIs. Black hollow rectangles indicate the presentation of each 50 ms tone in the tone train.

**Figure 5 genes-16-00672-f005:**
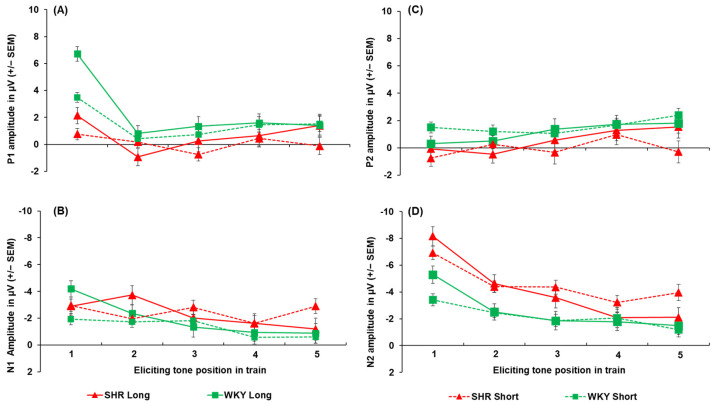
Peak amplitudes of P1 (**A**), N1 (**B**), P2 (**C**), and N2 (**D**) for both SHRs and WKY controls, not separated by sex, for both 1 s and 5 s (long) ITIs. Error bars represent the standard error of the mean (SEM).

## Data Availability

The dataset analyzed for this project is provided as a Appendix A.

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
