# Peer review of "Auditory Event-Related Potentials in Two Rat Models of Attention-Deficit Hyperactivity Disorder: Evidence of Automatic Attention Deficits in Spontaneously Hypertensive Rats but Not in Latrophilin-3 Knockout Rats"

_genes, 2025, doi:10.3390/genes16060672_

Round 1

Reviewer 1 Report

Comments and Suggestions for Authors

ADHD in rats – auditory potentials   Genes-3654753  
Reviewer Comments

This paper examines auditory evoked potentials on EEG in rats subjected to repetitive bursts (trains) of auditory stimuli, each burst being identical (five 50 msec tones equally-spaced over a 1.85 sec period), but with either a 1sec or 5 sec gap after the previous burst. Two different rat models of ADHD were compared (Lphn3 knock-out, and Spontaneously Hypertensive Rat -SHR), plus wild-type controls. The authors found that Lphn3 KO show no difference between the 1 sec and 5 sec interval from controls, but the SHR rats do, replicating similar previous observation in humans with ADHD. The SHR rat may therefore be the closer model to human ADHD.

Although rather long, the paper is very well written, though I suspect that the level of detail will probably restrict its accessibility and appreciation of its relevance to those already working in this field.   

For this reviewer, as a non-specialist in this field, my comments are necessarily aimed primarily at the presentation of the paper and clarity of description of the experimental design.  These are:

  1. Principle comment:
    Page 4; Line 161-3 : ‘…This passive auditory paradigm uses trains of 5 tones with inter-tone intervals (onset to onset) of 400 ms, each separated by either a 1-s (short) or 5-s (long) ITI…’
        Page 6: L247-9; L251-3: ‘… Following 2 min of resting EEG recording, the rat was presented with a series of tone trains, each preceded in randomized order by either a 1- or 5-s ITI, for a total of 320 trains/ITIs (160 long and 160 short).  …..   ….All tones were 50 ms in duration with 5-ms Gaussian onset and offset periods and 400-ms stimulus onset asynchronies (SOAs) within tone trains.’

The authors description of the timing of the auditory tone stimuli, is not as clear as it could be, and may lead to confusion for a reader.  In particular it is not immediately obvious whether the graphs plotted in Figs. 1 & 3 represent the averaged response to the first (or perhaps the 2nd) tone train, or to averaged responses to all tone trains coming after the 1s or 5s ITI, irrespective of whether they were the 2nd or 320th tone train presented to the animal.
Similarly, although Fig. 1 & 3, show diminution of response over the first 5 tone trains, it is not immediately clear whether the authors have analysed for likely diminution of response over the course of the 320 tone trains presented to each rat, as they only illustrate the first 5; -(or is the 320 the total number of tone trains across 64 or so rats ?)  

This can be largely overcome in a simple way by adding a very simple time-line diagram as an additional Figure.  Please see the attached pdf file for a suggested example of a Figure to achieve this (which would need to be in the main paper rather than as Supplementary)

  1. Figure 1 and all subsequent Figures.
    It is a pity that the authors have not chosen to present their graphs in colour, as this would better differentiate the 4 lines. Can the authors please consider this ?

3-8.  Other comments
The two comments above, and 6 other comments are given in the Table below.

No.

Page; Line

Current text

Suggested revision

Comment

3

P2; 68-69

‘..have been linked through genome-wide linkage studies to the risk of developing ADHD [10-13]…’

have been shown through genome-wide linkage studies to be associated with the presence of ADHD [10-13].

Is it linked with risk – ie. were the studies which showed this, longitudinal follow-up ones, or were they cross-section association studies ?.  Please check this from ref. 10-13

4

P2; 71

‘..and the susceptibility to developing childhood ADHD [13]…’

and the presence of childhood ADHD [13}

Please check ref. 13 as with the previous comment

5

P3; L118-9

And P4; L156-8

And P6; L287-9

‘..They consist of two obligatory components, the positive P1 and negative N1,..’

‘..To understand the influence of Lphn3 on inattention, we examined the auditory-evoked P1 and N1 and other auditory ERPs (P2 and N2) between Lphn3 KO rats and their wildtype (WT) control line [28]…’

‘...The following ERP peaks were defined: the most positive point 0–100 ms for P1, most negative point 50–150 ms for N1, most positive point 100–300 ms for P2, and most negative point 200–350 ms for N2…’

(???)

To understand the influence of Lphn3 on inattention, we examined the auditory-evoked P1 and N1 (occurring between 0-100ms after the initial stimulus) and other auditory ERPs (P2 and N2, occurring between 100-300ms after the initial stimulus)

(???)

between Lphn3 KO rats and their wildtype (WT) control line [28].

It is not clear what is the relationship between 1 and 2 in P1, N1 and P2, N2.  Ie. What does ‘other auditory ERPs’ mean’

They are not explained until lines 287-9 as:

‘The following ERP peaks were defined: the most positive point 0–100 ms for P1, most negative point 50–150 ms for N1, most positive point 100–300 ms for P2, and most negative point 200–350 ms for N2.’

A brief explanation needs to be put in when the terms P1, N1 and P2, N2 are first presented.

6 (1)

P4; L161-3

AND

P6: L247-9; L251-3

‘..This passive auditory paradigm uses trains of 5 tones with inter-tone intervals (onset to onset) of 400 ms, each separated by either a 1-s (short) or 5-s (long) ITI ..’

‘…Following 2 min of resting EEG recording, the rat was presented with a series of tone trains, each preceded in randomized order by either a 1- or 5-s ITI, for a total of 320 trains/ITIs (160 long and 160 short).  …..  

….All tones were 50 ms in duration with 5-ms

Gaussian onset and offset periods and 400-ms stimulus onset asynchronies (SOAs) within tone trains…’

See Main point 1 above

The authors descriptions of the timing of the auditory tone stimuli, is not as clear as it could be, and may lead to confusion for a reader.  In particular it is not immediately obvious whether the graphs plotted in Figs. 1 & 3 represent the averaged response to the first (or perhaps the 2nd) tone train, or to averaged responses to all tone trains coming after the 1s or 5s ITI, irrespective of whether they were the 2nd or 20th tone train presented to the animal. 

Similarly, although Fig. 1 & 3, show diminution of response over the first 5 tone trains, it is not immediately clear whether the authors have analysed for likely diminution of response over the course of the 320 tone trains presented to each rat, as they only illustrate the first 5; -(or is the 320 the total number of tone trains across 64 or so rats ?)  

This can be largely overcome in a simple way by adding a very simple time-line diagram as an additional Figure.  

A suggested example is attached as a pdf file.

7

P6; 293-4

‘..or the presence of significant outliers (i.e., ±3 SD) across several ERP components and measures were removed. ..’

Approx 17.5% of rats were removed (16/91).  The authors need to say how many outliers were there in the group who were removed – and discuss more fully, whether removing these outliers is valid.

8 (2)

P7; L333

Figure 1, and all subsequent Figures

See Main point 2 above

Figure 1 (and indeed all the Figures) would be easier to follow if the 4 different lines were in different colours. Can the authors consider this ?

Reviewer 2 Report

Comments and Suggestions for Authors

The study aimed to investigate the functional role of the Lphn3 gene. To this end, Lphn3 knockout (KO) rats were generated. These animals have shown impairments in behavioral phenotypes associated with Attention-Deficit/Hyperactivity Disorder (ADHD), such as deficits in working memory, increased impulsivity, and hyperactivity. The study specifically examined whether Lphn3 KO rats and Spontaneously Hypertensive Rats (SHRs) exhibit deficits in automatic attention, as measured by reductions in the P1 and/or N1 amplitudes in response to the first auditory tone presented after a 5-second intertrial interval (ITI). This approach aimed to determine whether the absence of Lphn3 expression leads to deficits in automatic attention and to further validate the SHR model in terms of its construct validity for ADHD research.

The results revealed differential patterns between the two animal models, which likely reflect distinct disruptions in dopamine and noradrenaline neurotransmission pathways—mechanisms that are thought to underlie core ADHD-related phenotypes.

Reviewer Comments:

  • Title: The title is informative and accurately reflects the content of the study.
  • Abstract: In the "Materials and Methods" section of the abstract, the study design and number of animals used are not clearly stated. The "Results" section should report the statistical significance of the findings and rely more heavily on quantitative data.
  • Keywords: The keywords should be reviewed and revised according to MeSH (Medical Subject Headings) terminology.
  • Introduction: The introduction effectively contextualizes the study of auditory evoked potentials and attention in ADHD, clearly stating the study's objectives. It draws upon relevant and up-to-date literature.
  • Materials and Methods:
    • Line 185 indicates that the study received approval from the University of Memphis Institutional Animal Care and Use Committee (IACUC), but the approval number is not provided.
    • The study design should be clearly described.
    • It is not stated whether a sample size calculation was performed.
    • The procedures are thoroughly described and supported by relevant literature, which justifies the methodological choices.
    • However, the statistical analysis methods used should also be detailed.
  • Results:
    • The results are presented in a clear, structured manner, supported by well-designed figures that enhance comprehension.
    • The accompanying explanations improve interpretation of the data presented in the figures.
  • Discussion:
    • The discussion reflects critically on the findings, and its structured organization facilitates reading.
    • The cited literature is highly relevant and strengthens the discussion.
    • The section on strengths and limitations (Section 4.3) offers a balanced and reasonable evaluation.
    • It would be valuable to expand on the potential clinical implications of the findings and to propose future lines of research based on the results.
  • Conclusion:
    • The conclusion should focus on summarizing the main findings and achievements of the study, rather than relying on a general summary of the literature.
